# Proteomics in Multiple Sclerosis: The Perspective of the Clinician

**DOI:** 10.3390/ijms23095162

**Published:** 2022-05-05

**Authors:** Dániel Sandi, Zsófia Kokas, Tamás Biernacki, Krisztina Bencsik, Péter Klivényi, László Vécsei

**Affiliations:** 1Department of Neurology, Albert Szent-Györgyi Faculty of Medicine and Clinical Centre, University of Szeged, H-6722 Szeged, Hungary; sandi.daniel@med.u-szeged.hu (D.S.); kokas.zsofia@med.u-szeged.hu (Z.K.); biernacki.tamas@med.u-szeged.hu (T.B.); bencsik.krisztina@med.u-szeged.hu (K.B.); klivenyi.peter@med.u-szeged.hu (P.K.); 2MTA-SZTE Neuroscience Research Group, University of Szeged, H-6722 Szeged, Hungary

**Keywords:** multiple sclerosis, proteomics, biomarker, pathogenesis, disease activity, progression, conversion, disease modifying therapy

## Abstract

Multiple sclerosis (MS) is the inflammatory demyelinating and neurodegenerative disease of the central nervous system (CNS) that affects approximately 2.8 million people worldwide. In the last decade, a new era was heralded in by a new phenotypic classification, a new diagnostic protocol and the first ever therapeutic guideline, making personalized medicine the aim of MS management. However, despite this great evolution, there are still many aspects of the disease that are unknown and need to be further researched. A hallmark of these research are molecular biomarkers that could help in the diagnosis, differential diagnosis, therapy and prognosis of the disease. Proteomics, a rapidly evolving discipline of molecular biology may fulfill this dire need for the discovery of molecular biomarkers. In this review, we aimed to give a comprehensive summary on the utility of proteomics in the field of MS research. We reviewed the published results of the method in case of the pathogenesis of the disease and for biomarkers of diagnosis, differential diagnosis, conversion of disease courses, disease activity, progression and immunological therapy. We found proteomics to be a highly effective emerging tool that has been providing important findings in the research of MS.

## 1. Introduction

Multiple sclerosis (MS) is a chronic, inflammatory, neurodegenerative disease of the central nervous system (CNS). It mainly affects young adults (20–45 years), it is predisposed toward women (female-to-male ratio 3:1), it is more common in the Caucasian race and it shows a north–south gradient (the highest prevalence can be measured in Scandinavia and Canada, while near the equator, the disease is almost unknown) [1,2,3,4,5,6,7,8,9]. It is considered to be a rare disease: approximately, there are 2,800,000 patients worldwide [10].

However, the significance of MS goes beyond the mere number of patients. Among the affected young adult/middle aged population, MS is the most common neurological cause of disability after epilepsy [11,12]. The disease seriously worsens the quality of life (QoL) of the patients in every measurable aspect: leading to restrictions in everyday life, causing unemployment, destroying social relations of the patients and even leading to divorce [13]. Moreover, the disease not only affects the QoL of the patients, but elevates their mortality risk threefold compared to the general population [14,15].

The aforementioned make it clear that early diagnosis, estimating the rate of aggression of the disease at the beginning and the earliest adequate therapy is of utmost importance. Thus, the identification of biomarkers (biological, radiological or psychological in nature) regarding either the pathogenesis, the activity, the progression of the disease or the monitoring of therapeutic effectivity is the hallmark in MS research.

In this comprehensive review, after the short summary of the pathogenesis, diagnosis and therapeutic considerations of the disease, we aim to present the usefulness of proteomics in the research of and the latest data on biomarkers identified by this method regarding MS, from the clinician’s point of view.

## 2. Multiple Sclerosis

### 2.1. The Pathogenesis of MS

There is no complete consensus on the pathological processes leading to MS due to conflicting data; however, the disease is considered to be a two-staged entity by most researchers. The more accepted view states that it begins with an autoimmune inflammatory response primarily against the oligodendrocytes in the CNS, leading to demyelination (inflammation), and, after a while, when the regenerative capacity of the oligodendrocytes is exhausted, the inflammatory processes are directed against the neurons themselves, leading to the permanent injury and dysfunction of the CNS (neurodegeneration) (Figure 1).

During the earlier inflammatory stage, two different (yet in effect, additive) processes can be identified. The first begins in the periphery when autoreactive lymphocytes enter the CNS through the damaged blood–brain barrier (BBB), recognizing antigens and initiating an inflammatory cascade by producing proinflammatory cytokines and recruiting more immune cells to the site [16]. These lymphocytes were initially thought to be almost exclusively CD4+ T cells; however, both the CD8+ T cells’ and the CD20+ B cells’ role has been shown to be mainly in the earliest, invisible stages of the disease [17,18,19]. The exact mechanism leading to the activation of those cells is not well understood, however. A number of processes have been suggested, such as the upset of balance between the proinflammatory and antiinflammatory T-regulatory cell phenotypes on the periphery or the impaired function of B-regulatory cells, natural killer cells and the antigen-presenting cells [20]. Recently, there has been a renewed interest in the potential role of the Epstein–Barr virus (EBV) infection as well, which—possibly through the process of molecular mimicry or by creating a self-amplifying inflammatory response—may commence this autoinflammatory cascade as new, convincing data emerged on the link between the infection and the development of MS [21,22]. Despite this, however, the hypothesis is yet to be thoroughly validated. Whichever process is the most important, the activation of all three type of lymphocytes results in the recruitment and activation of microglia and macrophages, highlighting the importance of the innate immune system in MS as well [23]. These activated mononuclear phagocytes’ roles are multilayered, sometimes even opposing each other. On the one hand, they are the effector cells of myelin destruction and removal, thus the primary cells responsible for the creation of the focal white matter lesions characteristic to the disease [24]. The level of myelin degradation products in these cells is even considered to be a measure of activity in these lesions [25]. On the other hand, however, several studies found that the “properly” conditioned microglia are able to promote remyelination as well. Induced by IL-4, microglia are able to promote the survival of oligodendrocytes and activate oligodendrocyte progenitors through several pathways (the brain-derived neurotrophic factor (BDNF), transforming growth factor beta (TGF-β), for example), thus playing a pivotal role in the process of remission [26,27].

Regarding the second inflammatory process, the key players are the CD20+ B cells. These cells can be found in the connective tissue in the CNS as well, mainly in the meninges and the Virchow-Robin spaces forming inflammatory aggregates, so-called “ectopic germinal centers”, there [28,29,30]. The activation of these cells leads to a diffuse damage in the normal-appearing white and grey matter, and the subpial cortical regions [29,30].

Through these inflammatory responses—mainly microglia being the source of the oxidative stress—the autoimmune process will inevitably lead to axonal dissection in the white matter and cortical and subcortical neuronal injury in the grey matter, evidently from the very beginning of the disease [31]. The most important consequent stressors brought on by inflammation seem to be glutamate toxicity and mitochondrial damage [32,33]. These processes disturb the highly intertwined and co-dependent metabolic pathways in the neurons and lead to energy deficit and Ca^2+^-overload, resulting in neuronal death [34]. Despite some compensatory mechanisms maybe taking place at the earlier stages, with time, maladaptive processes (e.g., ion channel redistribution) prevail and even accelerate neurodegeneration [35].

However, this view on the “primacy” of autoimmune inflammation has been challenged recently. Some promising data emerged implying that oligodendrocyte apoptosis and axonal damage, thus neurodegeneration, is the first step in the evolution of MS. This process reveals previously hidden antigens and activates the immune system, which leads to a secondary autoimmune response, initiating the whole “vicious circle” [36,37,38,39].

Whichever hypothesis is ultimately true, there are still many unanswered questions regarding the pathological processes leading to MS. If the disease is primarily an immune condition, what can be the initiator of the inflammatory response (e.g., the suggested molecular mimicry through EBV or another viral infection; a different type of infection or some other environmental stressor)? Why and how does the BBB get damaged and become unable to halt the invasion of the immune cells? What are the targeted antigens of the process? If the second hypothesis is true, what process leads to the oligodendrocyte apoptosis, and how and why does that happen? Again, how will the BBB be damaged to make the secondary immune response possible? Many more data are needed to give satisfactory answers to these questions. However, as new and new results accumulate, our understanding of the pathological processes behind the disease grows, and this will lead to further answers.

### 2.2. Clinical Phenotypes, Diagnosis and Therapy

Since the publication of the new phenotypic classification of the disease in 2014, the clinical courses of MS can be categorized into two basic types: the relapsing and progressive disease courses [40]. The relapsing disease course is the more common of the two, representing ~85% of the cases [41]. It consists of the “classical” relapsing-remitting (RRMS) course, which is characterized by the sudden onset of new or worsening neurological symptoms that subside within 3 months with or without sequalae (“relapses”), and stable periods (“remission”), and the clinically isolated syndrome (CIS), consisting of patients strongly suspected to have MS but not yet fulfilling all the diagnostic criteria of the disease [40]. Progressive MS consists of two subgroups: primary progressive (PPMS) and secondary progressive (SPMS) courses. SPMS is the “end-phase” of the relapsing disease course: after a period of time (10–20 years), the neurological state of the patients begins to gradually worsen and disability is continuously accumulated with or without any superimposed relapses [42]. In the case of PPMS, there is a continuous accumulation of disability from the very beginning of the disease and there is no preceding worsening–improving period [43].

Both basic clinical types can be further classified. In the case of the relapsing subtype, the disease can be active or not active, based on the clinical (new relapse) and radiological (new or enlarging T2 or T1 lesions on MRI) signs of activity [40]. Progressive MS has four subtypes (either PPMS or SPMS): active and progressive, active but not progressive, progressive but not active and not active and not progressive based on clinical and radiological activity and on the definition of progression (the confirmed disability worsening measured by the expanded disability status scale (EDSS) score) [40].

The accurate diagnosis of MS is based on proving that the disease is disseminated in space (DIS) and time (DIT) as well. Since the implementation of the new McDonald diagnostic criteria in 2017, the diagnosis has become faster without sacrificing any sensitivity or specificity [44]. Apart from two distinct clinical relapses, the criteria of DIS can be fulfilled if ≥2 T2-hyperintense lesions can be observed on the initial MRI scan on ≥2 characteristic (periventricular, juxtacortical, infratentorial, spinal) locations. If DIS is not fulfilled on the initial MRI scan, it can be fulfilled if a new lesion appears in a new location on any subsequent MRI scan. DIT can be confirmed either by the simultaneous presence of gadolinium (Gd+) enhancing and not enhancing lesions, or the appearance of new T2-hyperintense lesions on any subsequent MRI scan. The presence of oligoclonal gammopathy (OGP) in the cerebrospinal fluid (CSF) also proves DIT. Thus, the diagnosis of clinically definitive MS (CDMS) can be made immediately after the first clinical event with high accuracy requiring only one MRI scan and a lumbar puncture. Diagnosing PPMS is a bit more time-consuming however. In the case of a suspected PPMS patient, 1 year of continuous progression independent from relapse activity must be proven and another two of the following three criteria should be met:At least one T2 hyperintense lesion in 2 ≤ predilecting (periventricular, juxtacortical or cortical or infratentorial) regions;Two or more T2-hyperintense lesions in the spinal cord;OGP in the CSF.

In 2018, the first ever therapeutic guideline in MS was implemented by the European Committee for Treatment and Research in Multiple Sclerosis (ECTRIMS) and the European Academy of Neurology (EAN) in Europe [45,46]. This guideline’s recommendations covered many areas of MS care including a new approach on DMT initiation. Based on the guideline, the earlier, escalating treatment strategy was abandoned and a new strategy arose that defined the initial activity of the disease as one of the key points for choosing the appropriate therapy for the patient. Thus, based on the initial MRI lesion and relapse burden, three categories of disease activity could be distinguished: low disease activity, high disease activity and aggressive MS [47,48]. The available therapies are initiated based on these categories from the lower to the higher effectivity. Thus, classic platform therapies (interferon-β (IFN-β), glatiramer-acetate (GA)), teriflunomide (TER) and dimethyl-fumarate (DMF) are initiated in the case of low disease activity; natalizumab (NAT) and fingolimod (FNG) are the first choice in the case of high disease activity, while alemtuzumab, ocrelizumab and cladribine are advised to be commenced in the case of aggressive MS. Of course, in the case a patient shows any sign of disease activity, the utilized therapy should be escalated to a more effective treatment.

All these summarized data show that today, MS can be diagnosed earlier than before with good accuracy, the clinical course can be clearly defined and there is a clear guideline for the appropriate treatment on the clinical basis. However, there are still many aspects in which our knowledge is lacking. There are still many CIS or isolated optic neuritis cases, where an initial biomarker showing the potential conversion to CDMS would be invaluable. Moreover, there is no specific biomarker for the diagnosis of MS, which would greatly enhance the differential diagnostic potential. The diagnosis of SPMS is very difficult and is often retrospective as the clear boundary between RRMS and SPMS is undefined, thus, a biomarker for the conversion to SPMS would be extremely useful. Despite several attributes that set PPMS apart from relapsing MS, the diagnosis can only be made after at least 1 year of progression. Thus, patients would benefit greatly from a potential biomarker suggestive of PPMS at the very beginning of the disease. Last but not least, despite being able to give a good estimation on disease activity at the onset, highly accurate biomarkers would be very important for predicting the change in disease activity in the future. All in all, identifying specific molecular biomarkers for the aspects of MS are highly important and are considered hot topics in the current research. Proteomics, as a method, could be very effective in reaching these goals and providing us with these extremely important tools in the near future.

## 3. Proteomics

Since the beginning of the new millennia, biomedical sciences have taken a tremendous leap in progress. It began with the in-depth analysis of the human genome (e.g., the Human Genome Project) that provided detailed data about the structure, expression and function of genes, which gave birth to the science of genomics [49]. However, genomics alone was unable to explain the many possible variants of information stored in the human DNA. It was reasonably assumed that the intermediate and final products of a gene are more complex and are closer to the assessed specific function resulting in the specific observed phenotype than the gene itself [49,50]. This led to the “next step”, the birth of many “-omics”, all assessing a specific level of biological function (e.g.,: transcriptomics—assessing the mRNA structure and function; metabolomics—assessing the specific chemical signatures that are caused by a specific cellular process).

Proteomics is the branch of biomedical studies that specifically analyzes the whole protein content, their functions and interactions in a cell or a whole organism [49]. Despite there being many strategies by which it can be divided into subgroups, the most accepted division creates three main subdivisions [49]:Clinical proteomics: The discipline that analyzes the role of proteins as potential biomarkers in a given disease.Structural proteomics: This discipline aims to evaluate the three-dimensional structure of a protein in connection with its function in the cell.Functional proteomics: The main goal of this discipline is to analyze the interactions of a given protein with other proteins or other types of molecules in connection with its role in the complex (patho)physiological processes.

A plethora of technologies are in use under the umbrella-term proteomics such as the conventional analytical approaches, including chromatography-based techniques or enzyme-linked immunosorbent assay (ELISA), and the new, advanced technologies including mass spectrometry or the gel electrophoresis techniques. However, it is most practical to summarize these technologies based on their everyday application in research (Figure 2).

To purify biological samples, several chromatography-based techniques, such as ion exchange chromatography (IEC), size exclusion chromatography (SEC) and affinity chromatography, can be utilized [51,52]. The enzyme-linked immunosorbent (ELISA) and the Western blot are used to analyze selective proteins. These technologies are able to detect the presence of a given protein in the sample in extremely low concentrations, in some cases, even a single molecule [53,54,55]. However sensitive they may be, these technologies are restricted for the analysis of a few individual proteins. All these methods are useful in the case of the qualitative analysis of proteins. However, the need often arises for quantification and the separation of complex protein samples. For both goals, sodium dodecyl sulfate-polyacrylamide gel electrophoresis (SDS-PAGE), two-dimensional gel electrophoresis (2-DE), two-dimensional differential gel electrophoresis (2D-DIGE) or mass spectrometry can be utilized [51,56,57]. These techniques identify the proteins in the sample based on their molecular size, mass and charge; in the case of 2D-DIGE, with a dyeing procedure, the proteins can be visualized. For quantification, isotope-coded affinity tag (ICAT) labeling, stable isotope labeling with amino acids in cell culture (SILAC) and isobaric tag for relative and absolute quantitation (iTRAQ) techniques can also be utilized [51,58,59]. For the analysis of the three-dimensional structure of proteins, X-ray crystallography and nuclear magnetic resonance (NMR) spectroscopy are used [59,60]. Through continuously maintained and growing bioinformatics databases, the potential biological role and interaction of the assessed proteins can be assessed [51].

### 3.1. The Role of Proteomics in Biomarker Research for Neurological Conditions

Biomarkers were defined by the World Health Organization as “any substance, structure, or process that can be measured in the body or its products and influence or predict the incidence of outcome or disease” [61]. This broad definition allows any type of quantifiable attribute to become a biomarker; however, in molecular medicine, the molecules that are viewed as potential biomarkers are those that can reliably differentiate between a physiological and a pathophysiological process or predict or monitor a certain outcome in a given condition [30]. There is an abundance of molecular biomarkers routinely utilized in everyday medical practice: considering a routine blood test, C-reactive protein and procalcitonin levels are markers of inflammation, the liver transaminase levels measure the metabolic state of the liver, creatinine and urea levels measure kidney functions and the list goes on. However, when the CNS is regarded, it becomes clear that there is a shortage of diagnostic, monitoring or therapeutic biomarkers. The reason behind this is diverse, yet understandable. An ideal biomarker should be easily attained and reproduced by minimal or non-invasive means and still have good sensitivity, specificity and be affordable [61]. However, the CNS is a highly isolated and very sensitive system both structurally and functionally. This fact gives rise to several difficulties: (1) direct sampling usually requires a high level of invasiveness and may even pose some health risks (e.g., brain or spinal cord biopsy), so the possibility of routine use, not to mention routine repeated sampling, is limited at best. (2) Because of the isolated nature of the CNS, the possible molecular markers may not appear in the easily attainable samples (e.g., blood) or often do so in such low amounts that routine, repeated detection and quantification is very difficult and expensive. Despite these hardships, these obstacles can be overcome. With the advance in technology and innovation of proteomic analyses, more and more successes are being reported in the field of biomarker research of the nervous system.

Nowadays, there are two basic approaches for the quantitative proteomic analysis in neurological conditions [62]. The first is the so-called top-down analysis, which can be mass spectrometry-intensive or integrative [63]. The mass spectrometry-intensive technique utilizes a lower resolution gel-based separation of proteins first, then adds liquid chromatography–tandem mass spectrometry (LC-TMS) for identification [64,65]. Despite providing detailed information, it has a shortcoming: it mainly works in the 10–50 kDa range. The integrative approach begins with a 2DE separating proteins based on their charge, then utilizes the SDS-PAGE that identifies species by the mass/molecular weight, and, in the end, uses an LC-TMS (i.e., 2DE/LC-TMS) to identify potentially thousands of forms from any kind of biological sample [62,66,67,68]. On the other hand, in the case of the bottom-up strategy, protein extracts are first digested by protease enzymes resulting in a complex peptide mixture [69]. This mixture is then analyzed by LC-TMS [69]. Finally, through established online protein databases, the amino acid sequences potentially present in the sample are inferred. However, the shortcoming of this technique is that without extensive separate analyses, it loses all information concerning the different alternative forms of the proteins. Thus, despite the fact that the bottom-up approach may require less time and may be simpler, the top-down approach is more accurate in connection with the relevant alternative forms [70,71].

### 3.2. Biological Samples in Neurological Conditions

As was mentioned above, identifying biomarkers in the case of neurological diseases may have more difficulties than in other conditions due to the isolated and sensitive nature of the CNS. Despite this, the number of researches aiming to identify biomarkers for neurological conditions has exponentially grown in the last two decades. One of the key moments in this process is to choose the right type of sample for the analysis [72]. Here, we give a short summary of the potential advantages and the disadvantages of different biological samples used in proteomics studies of neurological diseases.

#### 3.2.1. Blood

Blood (both the plasma and the serum) is the single most “popular” biosample for molecular biomarker research in any medical discipline. It is easily collected by minimally invasive means and can be repeated as often as needed without causing any potential harm to the patient. It is abundant in different proteins and cells, including CNS specific molecules, thus makes an ideal sample for biomarker research [72,73]. Despite these advantages, however, there are several issues that could hinder the identification of a blood biomarker in neurological disease. Due to the isolation of the nervous system, CNS-related pathological proteins can usually be found in much smaller quantities in the blood than in the CSF or brain tissue, which is further diluted and cleared by the kidneys and the liver [74,75]. Furthermore, metabolic changes (e.g., protease activity during the process of getting from the CNS to the liver) can complicate or prevent the detection or quantification. Moreover, some highly abundant proteins (e.g., albumin) may mask the presence of other proteins in much smaller quantities, sometimes making detection impossible [72,74,75]. All in all, despite the shortcomings, blood is a logical and viable sample for proteomics research.

#### 3.2.2. CSF

The CSF is the special bodily fluid that provides mechanical and immunological protection to the CNS and plays an important role in its homeostasis. It is produced by the choroid plexus of the ventricles, circulates through the subarachnoid space and the ventricular system and is absorbed into the venous blood through the arachnoid granulations [76]. There is approximately 125–150 mL CSF present at any given time, while the mean daily production is approximately 500 mL [76]. Its composition is special, containing two magnitudes less protein than the plasma, is almost cell free (containing normally no red blood cells, and the normal number of leukocytes is 0–5/mm^3^) and the ion concentration differs from the plasma [77,78,79].

Due to its proximity to the CNS, its circulation and special composition, it may diffusely contain pathological proteins in the case of neurologic disorders and do so in relatively high concentration. Some of the proteins may only be found in the CSF due to its isolated nature. Moreover, due to its relatively lower concentration of albumin and other highly abundant proteins, proteins, even in small quantities, may not be masked in the CSF in contrast to blood samples [72]. All these attributes lead to special interest in the CSF regarding proteomics biomarker studies. However, there are some serious hindering issues as well. Obtaining CSF is a highly invasive procedure with some rarely occurring potential adverse events. This makes CSF sampling difficult, and repeated sampling almost impossible. Thus, despite its special state among the samples, the use of CSF is somewhat limited, albeit still very important in neuroproteomics research.

#### 3.2.3. Saliva

Saliva is the special fluid produced by multiple salivatory glands (e.g., parotid, submandibulary) and is the first digestive fluid of the gastrointestinal tract [80]. It has a special composition, the most abundant component in it being α-amylase, a digesting enzyme [81]. However, there are several other attributes of saliva, which have aroused interest in its analysis as an alternative sample in neurological conditions. First, it contains many types of proteins [82]. Second, sample collection is easy as it does not require any special medical training [83]. It requires less processing before sample analysis and a high number of proteins overlap with the proteins in blood samples [80,84]. There are some disadvantages, however, in its use. The most important of these, is that there are lot of proteins that do not filter into saliva from the blood or the CNS; thus, relevant biomarker candidates may not be present in it [85]. However, due to its collection and because processing is easy, saliva seems to be an acceptable alternative sample for neuroproteomics research in some cases.

#### 3.2.4. Tear

Tear is another alternative bodily fluid sample for proteomics research as its composition not only reflects the ocular tissue but the CNS as well. It is easily collectable by non-invasive means and repeated sampling poses no difficulties [86]. However, there are several obstacles in its routine use. First, it can be collected in very low amounts (5–10 µL) [86]. Second, there is a high interpersonal variability in its composition, and it is very sensitive to any effect before processing (e.g., light, irritants, etc.) [87,88,89,90]. Finally, the presence of highly abundant proteins may complicate its analysis, similar to blood [91]. All in all, however, tear is a viable option for neuroproteomics research on a limited scale.

#### 3.2.5. Urine

Urine has been more extensively studied than saliva or tear in the case of neurological disorders. It has some undeniable advantages, including easy, cheap and non-invasive sample collecting in large amounts that can be repeated any time [92]. Moreover, its handling, processing and storage have long had well-established protocols [92,93]. However, there are also some clear disadvantages in the case of proteomics analysis. First, metabolic variations are very common in urine [72]. Second, it has relatively low protein content; however, it is coupled with some high abundant proteins (e.g., albumin), and with the presence of relatively high levels of low molecular proteins [94]. These problems make the use of urine as a sample in neuroproteomics a limited option.

#### 3.2.6. CNS Tissue Sample

Brain or spinal cord tissue samples are natural choices for biomarker research when we consider CNS disorders. The direct and methodical analysis of the affected tissue from the affected site yields far more diverse and detailed results than other samples. However, for everyday utility, its viability is extremely limited. Sampling is extremely invasive and dangerous, making it a last resort effort in almost any disease, while repeated sampling is unthinkable in human subjects. The only way for performing routine evaluation is by post-mortem examinations in humans and dissection in animals. Thus, its utility, despite being highly important, is limited to animal models and basic research in the case of neurological disorders (with the exception of malignancies) [72].

### 3.3. Why Proteomics?

There are several attributes of proteomics that make it a better choice for biomarker research than other methods. Despite great advances in DNA sequencing and genetics as a whole, it was found that almost 40% of protein-coding genes lack experimental evidence at the protein level [95]. Moreover, the variance between the mRNA levels and protein levels can reach 20–30-fold differences [96]. Additionally, post-translational modification (acetylation or glycosylation for example) can very much alter the function of the proteins and its role in disease development or progression, which has no “sign” on the genomic or transcriptomic levels [97]. As proteins/peptides are the effector molecules in biological processes, these differences make the direct measurement of proteins inevitable.

We have elaborated on the several hardships in the identification of candidate protein biomarkers above—low abundance in easily collectable samples, the masking effect of resident proteins or their fast degradation by protease enzymes after sample collection, to name a few. However, new technologies are continuously being developed with success at overcoming these issues. For example, Luchini et al. reported the development of “smart” nanoparticles that can be immediately mixed with the collected sample and may perform chromatography and sequester the proteins in question away from albumin at the same time [98]. Despite this technology still being in the development phase and the costs being tremendous, the report does show that the technology of proteomics is evolving at a tremendous speed, and the obstacles of the present will be solved in the near future.

## 4. Results of Proteomic Studies in the Research of Multiple Sclerosis

The first proteomic studies were undertaken in the early 2000s; since then, their numbers have increased exponentially. These assessments encompass a wide range of strategic goals. Studies dedicated to the understanding of the pathological mechanism behind the development of the disease are probably the most abundant and were undertaken on both the classical animal models (experimental autoimmune encephalomyelitis (EAE) and cuprizone models) of MS and on human tissue samples. However, a plethora of assessments were aimed at finding molecular biomarkers for several aspects of the disease: diagnosis and differential diagnosis, forecasting disease activity and progression at the beginning of the disease, predicting conversion from one clinical course to another and also monitoring the therapeutic effect of different DMTs. In the following sections, we try to give an up-to-date summary on the results of these research.

### 4.1. Results in the Pathogenesis of MS

#### 4.1.1. Animal Models

Despite MS being exclusively a human condition, a number of animal models have been developed for the disease. The two most often utilized such models are the EAE and the cuprizone model. In the case of EAE, an inflammatory demyelination is induced in the CNS of the murine by either subjecting the animal to immunization with myelin antigens or by the direct transfer of myelin-specific T-lymphocytes [99]. The cuprizone model uses another approach, on the other hand, as it is a toxic demyelination model [100]. The animals are fed the copper-chelator *bis-cyclohexanone oxaldihydrazone* (aka cuprizone) which leads to demyelination [100]. However, 4 days after the cessation of the toxin, remyelination begins [101]. These methods are capable of modeling both the inflammatory and the neurodegenerative attributes of the disease and can also present the temporal course of the condition.

In the past two decades, several proteomics studies were conducted utilizing these animal models aiming to shed light on the pathological processes leading to MS. In 2010, Fazeli et al. assessed brain tissue samples from EAE mice compared to a control group [102]. They determined the up- or downregulation of 42 proteins with various known biological functions. One group were peptides involving ionic and neurotransmitter release (complexin-I, visinin-like protein (VILIP-1), calbindin 1 and 2), suggesting disturbances in both the excitatory and inhibitory synapses and potential pathways of motor function dysfunctions. Another group of proteins were members of the mitochondrial respiratory chain (complex I and IV), corroborating earlier results of histological studies suggesting the role of mitochondrial injury in lesion development [103]. Haptoglobin was also identified as a potential marker for BBB dysfunction. Protein phosphatase 1 catalytic subunit g1 and nucleophosmin 1 are thought to play a role in apoptosis induction. Finally, some structural proteins, including stathmin were also identified as being downregulated. Jastorff et al. searched for peptide markers for the process of neurodegeneration in 2009 in an EAE murine model [104]. Serum amyloid P component (SAP), a peptide associated with neurodegenerative and inflammatory conditions, was found in both the brain and spinal cord in elevated levels. Chloride intracellular channel protein 1, an important molecule in microglial activation, was also present in higher concentration. Stathmin-1 was identified in this study as well, among silent information regulator proteins (SIRTs) that are key molecules in myelin formation and remyelination. The dysregulation of these structural molecules may lead to neurodegeneration through disturbances in the formation and regeneration of myelin. Additionally in 2009, Jain et al. found evidence in EAE mice that proteolysis may be an important process in the pathogenesis of MS, implicating the role of several proteases (and their dysfunction) (α1-B-glycoprotein, β2-microglobulin, neurofilament light polypeptide and sulfated glycoprotein 1) in the pathogenesis of the disease [105]. Mikkat et al. found 26 peptides to have structural polymorphism in EAE-induced and EAE-resistant murine, including glia maturation factor-β, a protein inducing the expression of inflammatory cytokines in the CNS [106]. Hasan et al. generated a proteome-map of the brain of EAE mice in 2019 by identifying and quantifying more than 10,000 peptides and analyzing five regions: the brainstem, caudate nucleus, cerebellum, frontal cortex and hippocampus, independently. They found complement proteins, cell–cell adhesion proteins, proteins for T cell activation (Lcp1 and Pycard), transglutaminase, and microglial and monocyte markers to be upregulated in more than one brain region of EAE mouse [107]. Rosenling et al. measured a fivefold increase in total protein content of the CSF after EAE induction; however, they identified change in the amount of only 44 proteins, which discriminate between EAE and the control group. They found a non-synchronized onset of the disease, and that total protein content may indicate the disease developing earlier. They were able to identify proteins both in the early stages and in the “full-blown” stage of EAE. Most of the proteins identified were class I acute phase proteins, including α-1-inhibitor-3, fibrinogen, plasminogen, ceruloplasmin, hemopexin, haptoglobin, α-2-macroglobulin, α-1-antiproteinase, α-1-acid-glycoprotein (A1AG) and the components of the complement system [108]. A recent analysis conducted by Oveland et al. compared the proteome of the EAE and the cuprizone models using samples from the frontal cortex of the animals. As suspected, there were great differences between the two models. The results indicated that inflammatory, migratory and integrin signaling, microglia/macrophage activating pathways, and processes leading to astrocytosis and demyelination are the most affected in the cuprizone model, while acute phase associated proteins, granins and glutamate homeostasis proteins are the most affected in EAE [109]. The study also identified legumain, a lysosomal multifunctional protein that can exert situation dependent endopeptidase, carboxypeptidase and ligase activity to be highly upregulated in the cuprizone model, and indicated an association with inflammatory activity in MS lesions [109]. A top-down proteomic investigation of the cuprizone model found 43 proteins with “all-or-none” change. These proteins altered in the cortex have a role in diverse functions, such as axon growth, calcium signaling and energy metabolism among others [110]. Of these proteins, brain acid soluble protein 1, neurocalcin (a VILIP) and talin 1, an adhesion molecule, seem to be heavily downregulated, and, in the case of neurocalcin, may even disappear [110]. In 2012, Raphael et al. found that the proteins 14-3-3, glucose-6-phosphate isomerase (GPI), proteolipid protein (PLP1) and peroxiredoxin-1 (Prx1) are correlated significantly with disease progression in an EAE mouse model [111]. Another investigation subjected EAE brain endothelial cells to reactive oxygen species (ROS). They found that long-term ROS exposure led to the upregulation of the above mentioned Prx-1, an antioxidant, which in turn reduced cell death induced by ROS [112]. This seems to be a step in a global adaptive response in the case of ROS-induced injury. Another evaluation found that the decreased expression of CD47, an integrin-associated protein coincided with the disease onset in an EAE murine model [113]. MS is classically considered to be an immune disease initiated by the dysregulation of CD4+ Th1 and Th17 cells, thus its role in the pathogenesis of the disease is quite well established. However, it is surprising, that the renin–angiotensin–aldosterone system (RAAS) may have a role in organ-specific autoimmunity through the regulation of these cells. The proteomics evaluation in a very interesting study, however, showed that the members of the RAAS are upregulated in MS lesions; however, the blockade of these molecules (e.g., with angiotensin-converting enzyme (ACE) inhibitors) suppresses autoreactive Th1 and Th17 cells in EAE mice [114]. On the other hand, another important evaluation found plasma extracellular vesicles containing fibrinogen were able to induce CD8+ T cell-mediated spontaneous relapses in myelin oligodendrocyte glycoprotein (MOG)-EAE mice, which is not a classic clinical form of this model [115]. This result suggests the CD8+ T cells have a highly important role in the pathogenesis of the relapsing-remitting disease course. Another study identified kinin receptor B1 as a specific modulator that limits the recruitment of pathogenic T cells through the BBB into the CNS [116]. A study by Oveland et al. from 2017 evaluating the effect of 1,25-dihydroxy vitamin D in the cuprizone model suggested that it may have a positive effect on early remyelination through the regulation of proteins involved in calcium homeostasis and mitochondrial function [117]. The Daam2–VHL–Nedd4 axis was shown to be an important factor in remyelination after white matter injury in another study [118]. Montecchi et al. showed recently that, as well as oligodendrocytes, astrocytes may play a key role in the pathogenesis of MS, mainly in the processes behind neurodegeneration. They identified Rai, a key signaling molecule in the answers of astrocytes to the Th17 cell response to play an important role in neurotoxic effects through, as of yet, largely unknown pathways [119]. Its upregulation is suggested to be an important factor in influencing the proteasome–ubiquitin system, the main pathway of protein degradation in humans (and every mammalian species) [119]. Their latest assessment showed that Rai knock-out EAE murine astrocytes are able to survive oxidative stress, that the IL-17 dependent expression of hypoxia-inducible factors-1α is inhibited in these astrocytes and that extracellular vesicles of these cells produced as the result of IL-17-stimulus may provide defense against oxidative stress to other “bystander” cells [119].

#### 4.1.2. Human Tissue Samples

As animal models are only able to mimic the pathological processes leading to MS, studies on the pathogenesis of the disease naturally need to utilize human tissues. Of these, CNS samples would provide the most direct source for analysis; however, due to the insurmountable limitations of sampling on which we elaborated above, obtaining direct CNS samples is only confined to autopsies and very rare biopsies. Thus, the bulk of the data originate from research conducted on the CSF, blood and other alternative fluid samples, with all their limitations.

The rare studies using direct CNS sampling identified several candidate proteins in the pathogenesis of MS. In a pivotal proteomics study from 2008, several hundred proteins were identified that were unique to different pathological subtype of MS lesions as compared to healthy controls; however, no comparison was performed with normal-appearing white matter [120]. Further studies investigated different lesions with different levels of remyelination and in different locations. These assessments showed that plaques with ongoing remyelinating activities are abundant in peptides with much higher molecular weight than lesions without remyelinating activity [121]. Moreover, they revealed thymosin-β4 and hemoglobin interaction proteins (e.g., ATP synthase) to be involved in MS lesions [121,122]. Ninjurin-1, a protein implicated in the development of the peripheral nervous system, was shown to promote the migration of monocytes through the BBB into the CNS [123]. Another study examining the mitochondrial proteome from the cortex of MS patients identified cytochrome C oxidase subunit 5b (COX5b), a member of the Complex IV of the electron transport chain that can be dysregulated in MS patients [124]. Furthermore, hemoglobin-β and creatinine kinase B were also found to be upregulated in neurons of MS patients [124]. A recent analysis identified citrullination of myelin proteins, vimentin and CN37 to be of interest in the research of MS pathogenesis as it could be the consequence of an immune or inflammatory response leading to demyelination and the onset of the disease [125]. In 2020, Starost et al. identified CD4+ T cells to be mediators of impaired oligodendroglial differentiation that is caused by no intrinsic-factors, rather the proinflammatory environment of RRMS [126]. The in vitro analysis of B cells of MS patients showed the activation of cell survival and proliferation (mitogen-activated protein kinase (MAPK)), and proinflammatory (signal transducer and activator of transcription [STAT]) pathways [127]. In 2018, Nicaise et al. specifically evaluated the brain tissue of deceased progressive MS patients. They used senescent progenitor cells from the white matter lesions from the autopsy tissue and induced neural progenitor cells from patients with PPMS. They found these cells expressed senescent markers and that this expression could be reversed. They identified senescent progenitors to be the source of high-mobility group box-1, the protein blocking maturation and causing transcriptomic changes in oligodendrocytes, leading to incapability of remyelination [128].

Evaluations conducted on the CSF add to the above mentioned results. One such study from 2016 identified seven proteins potentially involved in the pathogenesis of MS, of which secretogranin-1 (Scg1) levels were elevated at the onset of MS as compared to healthy controls and RRMS patients [129]. Another evaluation from 2016 identified 26 proteins significantly dysregulated in MS samples compared to the controls, of which 9 were significantly less abundant, while 17 were significantly increased in case samples compared to controls [130]. A recent assessment found several hundred proteins discriminating between MS patients and healthy controls that were connected to processes such as inflammation, extracellular matrix organization, cell adhesion, immune response and neuron development [131]. These proteins included the highly researched neurofilament light chain (nFL) and Chitinase-3-like protein 1 (CHI3L1) and 2 (CHI3L2) [131]. Another recent evaluation found that mitochondrial proteins related to oxidative phosphorylation and sirtuin signaling pathways represent common pathways between white and grey matter lesions; however, their activation is opposite in direction [132]. Additionally, the downregulation of microfilament/cytoskeletal proteins in white matter lesions suggests impaired retrograde axonal transport [132]. As B cells evidently play an important role in the pathogenesis of MS, Hecker et al. evaluated B cell markers in their analysis. They found 54 immunoglobulin-type proteins that discriminated between MS and the control group, of which Epstein–Barr virus protein EBNA1 had the highest signals; however, it is important to note that RRMS and progressive MS groups overlapped [133]. A recent assessment found low levels of proteins playing key roles in neuronal development (complement proteins, semaphorin-7A, reelin, neural cell adhesion molecules, inter-α-trypsin inhibitor heavy chain H2, transforming growth factor β-1, follistatin-related protein 1, malate dehydrogenase 1 cytoplasmic, plasma retinol-binding protein, biotinidase, and transferrin) in MS patients compared to controls [134]. These results suggest an abnormally low oxidative capacity in MS patients leading to dysfunction in neural development at the very beginning of the disease [134].

In the case of serum/plasma or alternative fluid markers, an early study established that the pre-symptomatic MS cohort differed from the control group in 22 proteins’ expression [135]. Another evaluation identified ceruloplasmin, antithrombin III, clusterin, apolipoprotein E and complement C3 to be upregulated in MS patients as compared to a non-MS control group [136]. In 2016, a high quality study identified anoctamine-2 (ANO2) as a potential target antigen for the autoimmune response in MS, suggesting a possible ANO-2 subtype of the disease [137]. In an assessment profiling the autoantibody repertoire in MS patients, differences were found in the recognition frequencies of 51 antigens among MS patients [138]. Cvetko et al. found that the levels of core fucosylation and the high-mannose structures differed significantly between MS cases and controls, resulting in elevated proinflammatory potential [139]. One study using saliva samples identified 23 proteins that may distinguish between MS and control groups including subtypes of cystatin, statherin, antileukoproteinase and prolactin-inducible protein [140].

All the reviewed data, whether from animal models or human subjects, point to a mechanism behind the pathogenesis of MS that is far more diverse than we already know. Apart from the long-time suspected and (partly) proven role of immune cells and inflammatory pathways, several other key players were identified. The role of astrocytes, the molecular pathways involved in coagulation, mitochondrial functions, redox reactions, glial and neuronal differentiation, etc., all seem to be important in the development of the disease; thus, their further analysis is required for us to understand the whole picture. These studies have already marked several peptides as possible players for further in-depth reproduced analysis on large samples.

### 4.2. Results in Biomarker Studies

Despite the fact that having an understanding of the pathogenic mechanism behind the development of MS would be a tremendously important achievement, in everyday clinical practice, the diagnosis, differential diagnosis and the prognosis drive the therapeutic decisions of the neurologist. Thus, the identification of any potential biomarker that can aid a faster and more accurate diagnosis and differential diagnosis, the early accurate measurement of the temporal prognosis and the potential to monitor therapeutic effects could be considered a “holy grail” from the clinician’s point of view. In this section we try to summarize the results in biomarker studies aiming to find these candidate biomarkers that utilized proteomics.

#### 4.2.1. Diagnosis and Differential Diagnosis

Molecular biomarkers for the diagnosis of MS have long been of great interest to researchers. If we consider the diagnostic criteria of MS, we can realize, however, one such biomarker has been in use for many decades: OGP. Despite being sensitive yet not at all specific to MS, the presence of oligoclonal bands in the CSF have several attributes of importance: (a) it proves the presence of chronic inflammation in CNS; (b) it proves that the disease is disseminated in time; (c) its absence may aid in differential diagnosis (e.g., neuromyelitis optica spectrum disorder (NMOSD)). Despite these, the sensitivity and specificity prevents it from becoming more than a very helpful additional tool. Thus, molecular biomarkers are constantly researched and mainly involve the plasma/serum, the CSF and, in a smaller number of instances, other alternative fluids (urine, tear, saliva).

One of the first evaluations dedicated to molecular biomarkers for the diagnosis of MS was conducted in a pediatric population. This is not surprising as pediatric MS is quite rare and the other acquired forms of demyelination are more common than in adults, making differential diagnosis sometimes complicated [141]. This evaluation identified 12 proteins (A1AG, α-1-B-glycoprotein (A1BG), clusterin, gelosin, vitamin D-binding protein, hemopexin, kininogen, SAP component), which were upregulated in the MS cohort [142]. Notably, SAP and α-glycoprotein subforms were also indicated in earlier analyses as potential biomarkers [143,144]. In 2020, another pediatric evaluation by Solmaz et al. found higher levels of A1BG, complement factor B, plasminogen, α-2-antiplasmin and inter α-trypsin inhibitor heavy chain H2, and lower levels of centrosomal protein of 290 and F-box/LRR-repeat protein 17 in the sera of pediatric MS patients [145].

Several assessments were carried out on the adult population as well as searching for possible markers differentiating between MS and non-MS patients. Hassan et al. analyzed the CSF of MS patients and control subjects. They identified α-1-antichymotrypsin (SERPINA3), prostaglandin-H2 D-isomerase, desmoplakin and hornerin to be constantly upregulated in MS patients [146]. In 2012, Kroksveen et al. found 32 proteins in the CSF of RRMS patients that had levels that significantly differed from the control group. Among these proteins were immunoglobulin subunits, vitamin D-binding protein, apolipoprotein D, kallikrein-6, neuronal pentraxin receptor, Dickkopf-related protein 3 and contactin-1 [147]. In 2021, Shi et al. also compared MS patients’ CSF to non-MS controls. They identified the complement and coagulation cascades, Ras signaling pathway and PI3K-Akt signaling pathway as main components, but also insulin-like growth factor-binding protein 7 (IGFBP7), insulin-like growth factor 2 (IGF2) and somatostatin (SST) as potential differentiating markers [148]. A Danish evaluation carried out in 2021 found apolipoprotein C-I, apolipoprotein A-II, augurin, receptor-type tyrosine protein phosphatase-γ and trypsin-1 to be upregulated in the CSF of MS patients, regardless of the disease subtype as compared to healthy controls [149]. Another recent assessment of 2021 identified SERPINA3 and S100A4 as being increased in the CSF of MS patients compared to controls [150]. Berge et al. specifically evaluated the proteomes of CD4+ and CD8+ cells of MS patients against healthy controls. Their results highlighted the importance of the CD4+ T cell specific activation pathway [151]. When they specifically analyzed proteins of MS susceptibility genes, eight (LCK, GRAP2, CD5, ZC3HAV1, SAE1, EPPK1 and CD6 in CD4+ T cells and TNFAIP8 in CD8+T cells) were found to be dysregulated in the T cells of MS patients [151]. Jankovska et al. examined the CSF of women with newly diagnosed RRMS as compared to healthy controls. They found 69 proteins’ levels to be different between patients and controls, among them eosinophil-derived neurotoxin, Nogo receptor, ceruloplasmin and lysozyme-C [152]. An interesting study evaluating the urine samples of pregnant women identified changes in two proteins (trefoil factor 3 and lysosomal associated membrane protein 2) to not only discriminate between the third trimester and the postpartum period, but also between MS patients and the control group [153]. Another interesting evaluation in 2018 by Manconi et al. assessed salivatory samples of 49 MS patients compared to 54 control subjects. They found mono- and di-oxidized cystatin SN, mono- and di-oxidized cystatin S1, cystatin SA and mono-oxidized cystatin SA, mono-phosphorylated statherin levels to be lower, while antileukoproteinase, two proteoforms of prolactin-inducible protein, three proteoforms of P-C peptide, the SV1 fragment of statherin, two cystatin SN variants and a cystatin A variant to differentiate between MS patients and controls [140]. These proteins are linked to pathways of inflammation and neurodegeneration, further implying saliva samples may be good candidates in MS biomarker studies. In a very recent, high quality evaluation, Probert et al. investigated different CNS biomarkers that can help distinguish MS from non-MS controls as compared to OGP positivity. They found that the measurement of connective tissue growth factor/cysteine-rich protein/nephroblastoma overexpressed-5 (CCN5), von Willebrand factor (vWF) and glial fibrillary acidic protein (GFAP) increased the specificity and sensitivity level of the diagnosis as compared to solely identifying OGP [154]. CCN5 measurement increased both by 5%, while measurement of GFAP increased the specificity to 100% [154]. According to them, combining CCN5, vWF, GFAP with OGP is able to accurately discriminate between OGP positive MS and other neurological conditions with OGP positivity [154].

Biomarkers that can differentiate between the different clinical courses at the very beginning of the disease are also of special interest. The number of these types of evaluations is also rising. One of the first evaluations on this topic is from 2009 by Linker et al. They assessed both the animal model (EAE) in the later stages and compared the CSF of SPMS patients with RRMS and controls. They found GFAP levels to be upregulated in the spinal cord of EAE mice in the later stages of the disease and an increased level of GFAP specifically in the CSF of SPMS patients compared to both RRMS and the controls [155]. In 2010, Stoop et al. compared the CSF proteome of RRMS and PPMS patients. They found that the peptide composition of the two groups largely overlapped; however, they identified two proteins of interest for differentiation [156]. Jagged-1, a ligand for Notch signaling, was found to be three times less abundant in the case of PPMS patients; while vitamin D-binding protein was absent in PPMS patients, it was detected in more than half of the RRMS samples [156]. In 2011, Ingram et al. found factor H of the complement system to be elevated as compared to stable RRMS or CIS patients [157]. In 2016, Avsar et al. compared the CSF of CIS, RRMS and progressive MS patients. They identified RAAS, and the complement and coagulation pathways to be shared in all three subtypes of the disease [158]. However, they identified 151 proteins that were different. While they found the Notch pathway to be upregulated in progressive patients in concordance with its role in neurodegeneration, inflammatory pathways were more involved in the RRMS/CIS groups [158]. Both these evaluations suggest that the Notch pathway is a good candidate for further analysis in the case of progressive MS. In 2018, Martin et al. found four proteins to be differentially regulated among RRMS and progressive MS patients: tyrosine protein kinase receptor UFO, TIMP-1 and β-2-microglobulin levels showed an inverse correlation, while apolipoprotein C-II levels showed a positive correlation with CSF inflammation [159]. The aforementioned Danish evaluation resulting in proteins discriminating MS from controls from 2021, was also able to identify eight proteins that could distinguish between the clinical courses, of which SPMS was the most unique [149]. In 2014, Hinsinger et al. found that the levels of CHI3L1 in the CSF and the serum were higher in RRMS patients than in CIS, while CSF CHI3L2 levels were lower in RRMS than in progressive MS. The CHI3L1–CHI3L2 ratio in the CSF accurately discriminated between RRMS and progressive MS [160]. In the same year, an assessment from Italy found that secretogranin II and protein 7B2 were upregulated in RRMS patients compared to progressive MS patients, while fibrinogen and fibrinopeptide A were downregulated in CIS compared to progressive MS. Thymosin β4 levels were able to discriminate between CIS and RRMS patients [161]. SEPRINA3 levels were not only found to be discriminating MS from controls, but its levels were able to discriminate PPMS from RRMS as well [150]. In addition, IGFBP7 levels in the study of Shi et al. were not only different between MS and controls but were markedly elevated in both the CSF and sera of SPMS patients as compared to RRMS [148].

The last important role of diagnostic biomarkers would be their capacity to differentiate MS from other inflammatory, CNS or systemic disease that may mimic it. There are not many researches dedicated to this topic—we were able to identify only six, albeit their findings are highly important. The most important differential diagnostic problems are usually the so-called “related disorders” to MS: other acquired CNS demyelinating disease that can present similarly to MS, namely, NMOSD, Leber’s Hereditary Optic Neuropathy (LHON) and acute disseminated encephalomyelitis (ADEM). Other, systemic inflammatory conditions, such as Sjögren’s disease or systemic lupus erythematosus (SLE), may mimic MS as well, albeit for them to present as an isolated white matter neurological condition is considered extremely rare; thus, usually, the differential diagnosis causes fewer problems than in the case of NMOSD or LHON. Sometimes other non-inflammatory white matter diseases can present quite similarly to MS at the beginning; however, either the clinical or the radiological (or combined) presentations of these disorders usually have specific distinguishing markers that can help the diagnosis, not to mention the absence of evidence for inflammatory activity. All in all, it is not surprising that the bulk of the few researches on this topic deal with the differences between MS and NMOSD. The first evaluation on this topic using proteomics analysis is from 2012 conducted by Komori et al. [162]. They compared the CSF proteomics of MS patients with different clinical courses, seronegative and positive NMOSD, amyotrophic lateral sclerosis (ALS) and other inflammatory CNS conditions (aseptic meningitis, Guillan-Barré syndrome and chronic inflammatory demyelinating polyneuropathy (CIDP)). This study found that the proteomics pattern of NMOSD patients can be clearly differentiated from RRMS patients, and it is mainly due to differences during relapses. Moreover, it was observed that the differences between RRMS and PPMS were much more pronounced than between PPMS and amyotrophic lateral sclerosis (ALS) patients, another result pointing toward the highly neurodegenerative nature of PPMS. In 2014, Jiang et al. identified three proteins in the sera of MS and NMOSD patients that may aid in differentiating between the two conditions: Ig lambda chain, keratin 83 and haptoglobin [163]. The Ig lambda chain was only found in MS patients and the expression of keratin 83 was significantly elevated in the MS group, while the expression of haptoglobin was more than twofold elevated in the NMOSD group [163]. Lee et al., in 2016, compared the exosomal proteome of the CSF in MS and NMOSD patients. They found GFAP to be rather specific to NMOSD, its level being elevated as compared to both MS and controls samples, while fibronectin seemed to be more specific to MS [164]. The finding on GFAP seems a bit surprising considering other evaluations; however, it probably shows that GFAP may be an even more important marker in NMOSD than in MS [154,164]. Two interesting assessments evaluated the urine proteome trying to differentiate between MS and NMOSD. In 2015, a Danish assessment found that a 3-protein profile of the NMOSD patients discriminated their samples from the control group, while a 6-protein profile did the same against MS samples, while an 11-protein profile discriminated MS from the control group [165]. In 2016, Gebregiworgis et al. analyzed the urine samples of both MS and NMOSD patients. They were able to identify 27 metabolites altered differently in the urine of MS and NMOSD patients. These peptides were associated with ketone body, amino acid, propionate and pyruvate metabolism, and tricarboxylic acid cycle and glycolysis [93]. The one remaining evaluation using proteomics for the differential diagnosis of MS was conducted in 2008 and compared MS patients to LHON. They found a more than threefold increase in the level of apolipoprotein A-IV in the CSF of LHON patients as compared to MS, suggesting it may be a possible differentiating marker between the two conditions [166].

All in all, a high number of potential biomarkers for the diagnosis and the differential diagnosis of MS have been found to date, but most of them are yet to be validated on any level. The biomarker candidates with the greatest potential identified by proteomics, however, seem to be GFAP, CHI3L proteins and SERPINA3 at the moment, based on the reviewed literature.

#### 4.2.2. Conversion from CIS to RRMS and from RRMS to SPMS

Aside from the pathogenesis and the diagnosis of MS, another aspect has become important: to find biomarkers that can potentially either predict, or at least pinpoint, the time of conversion from one clinical course to another. Data on these type of biological markers evaluated with proteomics are scarce however. Here, we summarize their results.

The first proteomic analysis dedicated to identifying biomarkers of conversion from CIS to RRMS is from 2009. In this research, the authors compared the CSF of eight originally CIS patients who remained CIS after 2 years to eight patients who converted into RRMS during that period. They found one protein to be upregulated (serin peptidase inhibitor) and eight proteins to be downregulated (A1BG, fetuin-A, apolipoprotein A-IV, haptoglobin, human Zinc-α-2-glycoprotein, retinol-binding protein, superoxide dismutase 1, transferrin) in patients who converted into RRMS as compared to those who remained in the CIS stage [167]. However, it is important to note that by the time the research was conducted, the diagnostic criteria were different than they are now; thus, a portion of the patients would already be considered RRMS today instead of CIS based on the 2017 revision of the McDonald criteria [44,168]. Thus, these results should be interpreted with some caution. Next, in 2010, Comabella and colleagues used a similar design on a larger population (30 converting and 30 non-converting CIS patients), also utilizing CSF samples. They were able to identify three proteins at the first phase (ceruloplasmin, vitamin D-binding protein and CHI3L1); in the second phase, however, they could only validate CHI3L1 to have a significantly elevated level in the converters [169]. In 2015, Hinsinger et al. replicated the role of CHI3L1 on their own sample [160]. In 2016, another investigation further strengthened these previous data: Borras and colleagues were able to build a statistical model for conversion from CIS to RRMS based on the CSF levels of CHI3L1 and ala-β-his-dipeptidase [170]. In 2019, another CSF evaluation using a 5-year follow-up period identified four candidate proteins (CHI3L1, homeobox protein Hox-B3, tumor necrosis factor (TNF) receptor superfamily member 21 and iduronate 2-sulfatase) for conversion with LC-MS/MS; however, the only ELISA-validated protein was Hox-B3, making it another possible biomarker for conversion [171]. In 2014, Cantó et al. found that ala-β-his-dipeptidase and semaphorin 7A levels were elevated in the CSF of converting patients, further strengthening the possibility of ala-β-his-dipeptidase being a biomarker as well [172]. Pavelek et al. assessed RRMS and CIS patients compared to control groups. They identified 26 dysregulated proteins, of which 3 (Ig-γ-1 chain C region, Ig heavy chain V-III region BRO and Ig κ chain C region) were only present in the RRMS patients, making them a possible early biomarker for conversion [130]. Probert et al. reported the results of their up to 10-years follow-up study, where they assessed the possible markers for CIS to RRMS converters. They found several dysregulated proteins among other predictive factors of which DNA repair protein XRCC1, dynein light chain Tctex-TYPE 1 and natural cytotoxicity triggering receptor 1 had the most predictive power [173].

An interesting assessment from 2017 identified four neuro-axonal proteins (amyloid-like protein 1, contactin 1, contactin 2 and neuronal cell adhesion molecule) to be far less abundant in the CSF of patients during their first demyelinating attack than those of the control group [174]. However, there were no differences in the levels of these proteins between converters and non-converters [174]. The reason behind this is not well understood but may point to different processes leading to the first event and to conversion. A previous assessment from 2014 went back a step even further. They assessed the sera of MS patients sampled prior to showing any symptoms of the disease and matched it to a control group [135]. They found several proteins (including complement factor I, serum amyloid P component, fibronectin and biotinidase) to be dysregulated [135]. The affected pathways mainly had connections to inflammation, lipid and protein metabolism and cell communication.

In the case of conversion to SPMS, data obtained via proteomics are lacking. The only study we were able to identify regarding the subject is from 2020 by Huang et al. They found Eotaxin-1 (CCL11) to be increased in both the CSF and the plasma of SPMS patients as compared to RRMS patients [175]. It was also shown that this protein was connected to disease duration, with a mean approximate yearly increase of 1.1% and 1.9% in plasma and CSF [175].

Naturally, many more studies are needed for further validations; however, there are already a couple of good candidate biomarkers that could predict conversion from CIS to RRMS. Among these, CHI3L1 has the most robust backing data, elevating it to be the best candidate for a biomarker of CIS to RRMS conversion. In the case of conversion to SPMS, however, data are sporadic at best, necessitating further research.

#### 4.2.3. Disease Activity

Biomarkers that can potentially measure or predict the activity or aggression of the disease would be invaluable in clinical practice. This kind of biomarker could have a direct influence on therapeutic decision making helping to choose the most appropriate therapy for the patient at the onset of MS.

However desirable these biomarkers would be, data available on the topic are quite scarce, as we were able to identify only four articles. In 2010 Sawai et al. analyzed the sera of 31 RRMS patients. They found in their assessment the complement C4a fragment to be elevated during relapses and decreased in remission, thus representing activity of the disease [176]. In 2011, Li et al. identified complement C4b to be elevated in the CSF of active RRMS patients [177]. In 2012, Füvesi et al. analyzed the CSF of a fulminant MS patient and compared it to the sample of an RRMS patient and a control. They found 30 proteins to be increased in MS samples as compared to the control patient; however, they identified that only seven peptides (Ig-Κ and γ-1 chain C region, complement C4-A, fibrinogen-β-chain, serum amyloid A protein, neural cell adhesion molecule 1 and β-2-glycoprotein 1) were elevated in the fulminant case and not in the RRMS control, representing potential markers of interest for high disease activity [178]. Huang et al., in 2020, identified chemokine ligand 20 (CCL20) to be associated with the multiple sclerosis severity score (MSSS) [175].

Despite the scarcity of evaluations, three out of four assessments identified peptides of the complement system, the C4 protein in particular, to be associated with a more severe disease course in both the CSF and sera of the patients. These results suggest C4 to be a good candidate for being a biomarker of disease activity; however, further studies on a larger population are needed for validation.

#### 4.2.4. Disease Progression

As well as disease activity, the other hallmark process defining MS is progression: the accumulation of disability over time as measured by the EDSS score. Similarly to activity, any biomarker predicting or monitoring progression would be of utmost importance in therapeutic decision making and for the prognosis of the individual patient.

Again, similarly to disease activity, research using proteomic analysis to discover biomarkers for MS progression is scarce; however, their results are encouraging. In 2015, Kroksveen et al. found secretogranin-1 to be elevated in the CSF of CIS patients as compared to RRMS and controls as was reviewed above. However, when this result is combined with an earlier study on patients with an average disease course of 16 years that found secretogranin-1 levels to be decreased in that population, this suggests secretogranin-1 to also be a possible marker of disease progression in RRMS [129,179]. In addition, in 2015, a high quality assessment compared the serum proteome of “benign” MS patients and “aggressive” MS patients. Though there is more than one definition for both entities, they utilized the definition of EDSS score ≤ 3 points after ≥20 years of disease duration for “benign” MS and EDSS score ≥ 6 points within 10 years of the disease onset for “aggressive” MS [30,180,181]. They identified 11 proteins, of which 7 had increased levels in the aggressive phenotype (thrombospondin-1, platelet basic protein, insulin-like growth factor-binding protein 3, cholinesterase, protein S100-A9, platelet glycoprotein V and ficolin-2) and 4 in the benign phenotype (leucine-rich α-2-glycoprotein, corticosteroid-binding globulin, iter-α-trypsin inhibitor heavy chain H4 and lipopolysaccharide-binding protein) [180]. These proteins are related to inflammation, opsonization and complement activation [180]. This evaluation showed that there can be marked differences in the protein profile between the different levels of severity. In the very recent research of Comabella et al., the authors were able to validate CHI3L2 from 10 proteins to be markedly elevated among progressive patients with higher EDSS scores, thus discriminating with good sensitivity and acceptable specificity (90% and 63%, respectively) between high and low disability progression in patients with progressive MS [182]. In 2019, Malekzadeh and colleagues evaluated baseline protein biomarkers for both EDSS and MRI progression after a 4-year follow-up period. They found possible protein markers E-cadherin, checkpoint kinase 1, lectin, galactoside-binding, soluble 8, TNF receptor superfamily member 13B, repulsive guidance molecule family member A and macrophage inflammatory protein 1-α were all associated with EDSS progression [183]. Annualized percentage brain volume changes correlated with complement C3, fibroblast growth factor 9 and euchromatic histone lysine methyltransferase 2 [183]. These proteins are associated with the pathways of cell–cell and cell–extracellular communication, adherence, immune system communication and immune system activation. However, it is important to note they could not find overlapping markers for both parameters. In 2016, Lewin et al. assessed the association of brain atrophy during a 2-year follow-up in 140 SPMS patients with the serum proteome of the patients. They were able to find association between brain atrophy and free serum α- and β-hemoglobins [184]. They hypothesized that through the damaged BBB, hemoglobin and its breakdown products are able to enter the CNS and contribute to neurodegeneration and, consequently, brain atrophy [184]. In 2019, Magliozzi et al. found significantly higher levels of sCD163, free hemoglobin, haptoglobin and fibrinogen in the CSF of MS patients with high cortical lesion load at diagnosis. On the other hand, sCD14 levels of the CSF were higher in MS patients with low cortical lesion load. These results imply that these proteins might be possible early markers for early cortical damage [185]. A recent analysis from Sarkar et al. found that lower levels of mitochondrial fumarate hydratase are negatively associated with the duration of the progressive phase of MS [186]. These findings, as DMF is a fumaric acid ester, have some therapeutic implications as well. An interesting study emerged in 2018, however. Bridel et al. analyzed the plasma proteome of 67 RRMS patients at the time of diagnosis and aimed to identify any prognostic markers for long-term clinical outcomes in a 11-year-long follow-up period. Interestingly, they identified no association between the plasma proteome (1310 proteins in total) at diagnosis and clinical, cognitive or MRI outcomes after 11 years [187]. They concluded that for this type of evaluation, CSF samples might be better suited [187].

An interesting subset of research should be noted here as well, albeit concerning disease progression in a looser sense. There are some interesting epidemiological studies implying that ischemic events are more prevalent and result in a higher risk of mortality of MS patients, mainly in the progressive phases of the disease [188,189,190,191]. Some studies connected this phenomenon to disturbances in platelet functions and the thrombotic pathways [192]. Two research were dedicated to this topic utilizing proteomics approaches. In 2016, Bijak et al. found that four proteins’ (fibrinogen, α-2 macroglobulin, septin-14 and tubulin β-1 chain) levels directly connected to the thrombotic cascade were elevated in the platelets of SPMS patients [193]. The same group in 2021 found increased levels of P-selectin expression in the blood samples of SPMS patients, which is a known activation marker of platelets [194]. Both studies indicate the hyperactivity of platelets and the coagulation cascade in SPMS, supporting the higher risk of thrombotic events in the late stages of MS.

All in all, despite the growing amount of evidence, there is only an enlarging number of candidate molecules for MS progression and no definitive markers so far. However, these promising results show the capability of proteomics in finding new and new possible biomarkers in this field.

#### 4.2.5. Monitoring Therapeutic Effectiveness

The regular clinical and radiological follow-up of MS patients is of utmost importance and the core step in the quality management of the disease. However, despite these tools, we can usually only predict the effectiveness of the chosen DMT with a certain level of confidence at initiation, and we are unable to monitor it during treatment aside from radiological and clinical measures, which, by definition, mean there is a new clinical attack, confirmed progression or new MRI lesions, all of which we aim to avoid. Thus, molecular biomarkers that can predict at the beginning or prove during the course of treatment the effectivity of the chosen therapy would be of tremendous help to the clinicians in everyday practice.

Research using proteomics are not widespread in this topic either. However, there are some promising data that have emerged showing candidate markers to monitor our existing DMTs more precisely than we are able today. The first such investigation dates back to 2007 and concerns IFN-β treatment. Alexander et al. found 14-3-3, metavinculin, myosin-9, plasminogen, reticulocalbin-2 and-3, ribonuclease/angiogenin inhibitor 1, annexin A1, tropomyosin and Ras-related protein Rap-1A to be elevated in endothelial cells exposed to RRMS sera, and that the level of these proteins may decrease after IFN-β treatment [195]. In 2013, Stoop et al. assessed the effect of NAT treatment on the CSF proteome of MS patients. They found the levels of Ig mu chain C region, haptoglobin and CHI3L1 to be significantly decreased after 1 year of NAT treatment [196]. In 2019, Bedri et al. found nine proteins, with phosphatidylethanolamine-binding protein 1 (PEBP1) and reticulon 3 (RTN3) levels being the most significant in the sera of 44 patients to decrease in RRMS patients during NAT treatment and to validate this decrease in the case of PEBP1 on an independent cohort [197]. They also found that the levels of these proteins remained stable after the change to FNG treatment [197]. In 2016, Blewett et al. found that DMF was able to block the activation of both mouse and human T cells and identified—through assessing DMF-sensitive cysteine residues—protein kinase C θ (PKCθ) to be a key mediator in this process [198]. In 2021, Lozinski et al. measured the impact of exercising on the proteome of the spinal cord of murine with demyelination. They found 171 dysregulated proteins in the spinal cord and 25 dysregulated proteins in the sera after exercise, by which they were able to validate the decrease in the amount of two (connexin-32 and myelin basic protein (MBP)) in the lesion environment [199].

### 4.3. Therapy

There is another important aspect of MS not yet elaborated on in this article. Apart from the pathogenesis and the diagnostic or prognostic markers of the disease, proteomics may be used to uncover potential new targets for therapy.

In 2008, Han et al. assessed the brain tissue of deceased MS patients, analyzing the different type of plaques (acute, chronic active and chronic). Their analysis identified tissue factor and protein C inhibitor, both members of the coagulation cascade, are dysregulated in chronic active plaques. After that, they treated EAE mice with hirudin or recombinant activated protein C, that resulted in the decrease in proliferation and inhibition of Th1 and Th17 cytokines in the mice [120]. In 2015, Yun et al. found peroxiredoxin 6 (PRDX6) to be elevated in the spine of EAE mice and that PRDX6 transgenic mice presented with a significantly less aggressive disease course [200]. This evaluation might point toward PRDX6 being a possible therapeutic marker, or even a potential target for immunotherapy. A recent high quality research from Bernardo-Faura et al., recruiting 195 MS patients and 60 control subjects, yielded interesting results. The utilized DMTs were IFN-β, GA, NAT, FNG and the experimental drug EGCG. By developing a literature- and database-based prior knowledge signaling network (PKN) they aimed to find effective combination therapies based on the differences of signaling networks between plasma-derived immune cells of treated MS patients and controls. The PKN included molecules of interferon response pathways, B and T cell receptor signaling, cellular survival and apoptosis, inflammation, lipid signaling, innate immunity and multidrug response (MDR) genes. The model found the TGF-β-activated kinase 1 (TAK1) pathway, an important player of inflammatory and immune pathways, to be activated under all DMT treatment. After, utilizing the EAE mice model, based on these findings, they were able to validate the potential efficacy of FNG + TAK1-inhibitor combination therapy [201].

Despite there being only a few assessments regarding the identification of new therapeutic targets for MS using the proteomic approach, their results point toward new and exciting pathways and molecular targets to consider. They also prove that proteomics may be a highly effective method for undertaking such research.

## 5. Conclusions

In the last decade, the management of the disease has been rapidly evolving, and as of today we have reached the state where our (the clinical neurologist’s) aim is the implementation of personalized medicine. The latest diagnostic guideline and the new phenotypic classification helps us in the fast and sensitive diagnosis of the disease and the identification of the clinical courses, while the therapeutic guideline gives clear recommendations for the use of available DMTs and the criteria of an MS center has already been defined [40,44,45,46,202]. Thus, the only acceptable goal in MS care is now the fastest initiation of the therapy best suited for the patient with an MS center up to the task providing this need [203]. However, the goal is not only to slow down the disease but to effectively halt the activity and the progression and reach the so-called “No Evidence of Disease Activity” (NEDA) state, effectively meaning the patient experiences no further sign of the disease [204,205,206,207]. This goal may seem far-reaching, but it is attainable and highly important in the case of MS patients; we usually treat women and men in their prime, who are among the bulk of the workforce and at the age of starting their own family and raising their children. Thus, apart from the medical viewpoint, their health is of utmost importance for the whole society as well (from both the economic and the social point of view). Therefore, it is not surprising that governments tend to finance their complex management despite the very high costs of these immunological therapies [208,209,210].

However fast evolving the knowledge about MS is, there are still many aspects of the disease that are in need of new discoveries and innovations. The pathogenesis of the disease is thus far largely unknown, and there are still many aspects of the disease that we are only able to identify or monitor retrospectively (e.g., SPMS) or by the re-appearing clinical or MRI activity, which we would like to avoid. Therefore, from the clinical neurologist’s point of view, there is a dire and growing need for validated biomarkers by which we can predict and effectively monitor the disease in the long term. Thence, methods that can help to identify such markers will always be in the focus of research.

Proteomics is a continuously developing discipline of molecular sciences that greatly enhances the possibility of identifying proteins relevant in the neurological sciences, thus meeting the above mentioned requirements. It has been producing more and more data that have led to important new discoveries in the pathogenesis of MS and yielded candidate biomarkers—some with already robust possibility for clinical utility in the future—for many aspects of the disease (summarized in Table 1).

In our review, we aimed to give a summary of the importance of this methodology and a comprehensive review on its results that are, or may be relevant, to the everyday clinical practice in the near future, highlighting the more promising or surprising findings. As this field is under constant exponential evolution, we believe that many more data will emerge in the near future that will further support the findings reviewed here and herald a new and exciting era of personalized medicine in MS.

## Figures and Tables

**Figure 1 ijms-23-05162-f001:**
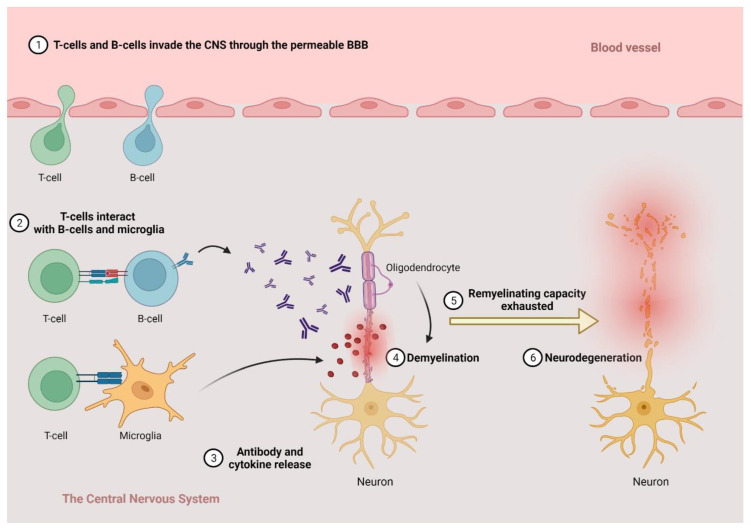
The schematic interpretation of the pathological processes behind multiple sclerosis. Created with BioRender.com—accessed on 25 April 2022.

**Figure 2 ijms-23-05162-f002:**
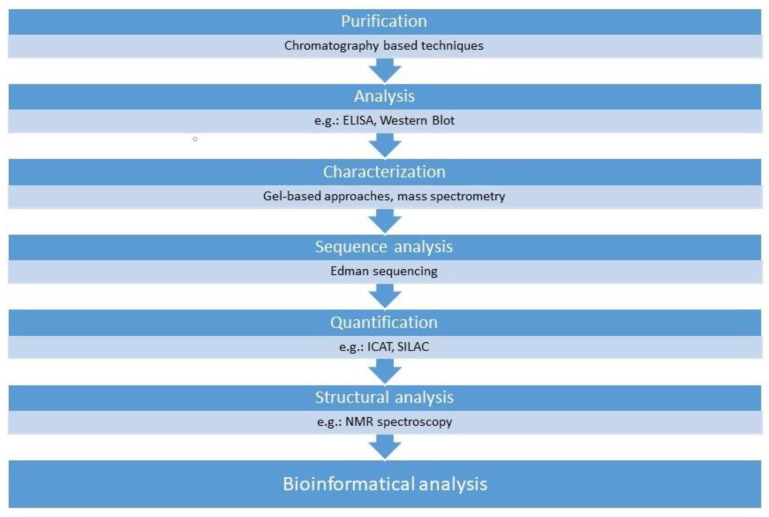
Application of different proteomics technologies. ELISA, enzyme-linked immunosorbent assay (ELISA), ICAT, Isotope-coded affinity tag labeling; SILAC, stable isotope labeling with amino acids in cell culture; NMR spectroscopy, nuclear magnetic resonance spectroscopy.

**Table 1 ijms-23-05162-t001:** Biomarker candidates of the different aspects of multiple sclerosis: peptides with at least two different proteomic studies implying their role in the disease.

Candidate Protein	Pathogenesis—Animal Model	Pathogenesis—Human Tissue	Diagnosis	Differentiating between Clinical Courses	Differential Diagnosis	Conversion from CIS to RRMS	Disease Activity	Progression	Monitoring Therapy	Sample Type(s)
*ala-β-his-dipeptidase* [170,172]	-	-	-	-	-	2	-	-	-	CSF
*Apo-A* [149,166,167]	-	-	1	-	1	1	-	-	-	CSF
*Apo-C* [149,159]	-	-	1	1	-	-	-	-	-	CSF
*Biotinidase* [134,135]	-	1	-	-	-	1	-	-	-	CSF, serum
*Ceruloplasmin* [108,136,152,169]	1	1	1	-	-	1	-	-	-	EAE CSF, CSF, serum
*CHI3L1* [131,160,169,170,196]	-	1	-	1	-	2	-	-	1	CSF
*CHI3L2* [131,160,182]	-	1	-	1	-	-	-	1	-	CSF
*Complement C3* [107,108,134,136,183]	2	2	-	-	-	-	-	1	-	EAE brain tissue, EAE CSF, CSF, serum
*Complement C4* [107,108,134,176,177,178]	2	1	-	-	-	-	3	-	-	EAE brain tissue, EAE CSF, CSF, serum
*Fibrinogen* [108,115,161,178,185,193]	2	-	-	1	-	-	1	2	-	EAE immune cells, EAE CSF, CSF, human platelets
*GFAP* [154,155,164]	-	-	1	1	2	-	-	-	-	EAE spinal cord tissue, CSF
*Haptoglobin* [103,108,163,167,185,196]	2	-	-	-	1	1	-	1	1	EAE brain tissue, EAE CSF, CSF, serum
*Plasminogen* [108,145,195]	1	-	1	-	-	-	-	-	1	EAE CSF, serum, human endothelial cells
*SAP* [104,142,143,144]	1	-	3	-	-	-	-	-	-	EAE brain tissue, plasma
*Secretogranin-I and II* [129,161,179]	-	1	-	1	-	-	-	1	-	CSF
*SERPINA3* [146,150]	-	-	2	1	-	-	-	-	-	CSF
*Stathmin-1* [103,104]	2	-	-	-	-	-	-	-	-	EAE brain and spinal cord
*Thymosin-β4* [121,122,161]	-	2	-	1	-	-	-	-	-	Human brain tissue, CSF
*Transferrin* [134,167]	-	-	1	-	-	1	-	-	-	CSF
*VILIP-1* [102,110]	2	-	-	-	-	-	-	-	-	EAE and cuprizone brain tissue
*Vitamin D-binding protein* [142,147,156]	-	-	1	1	-	1	-	-	-	CSF, serum
*α-2-macroglobulin* [101,193]	1	-	-	-	-	-	-	1	-	EAE CSF, human platelets
*α-glycoproteins* [105,108,142,143,144,145,167]	1	-	4	-	-	1	-	-	-	EAE brain tissue, EAE CSF, CSF, serum

## Data Availability

Not applicable.

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
