# Peer review of "Proteomics in Multiple Sclerosis: The Perspective of the Clinician"

_ijms, 2022, doi:10.3390/ijms23095162_

Round 1
Reviewer 1 Report
Authors provide here a complete review on the subject. The review is well documented and really informative.
Two sub-section have the same number 2.1
I would suggest to talk about the innate immunity in the first 2.1 sub-section since the innate immunity is deeply involved in the MS pathology and pathogenesis. Microglia is both detrimental and beneficial, it would be interesting to add few lines in the 2.1 sub-section. DOI: 10.3389/fncel.2020.00284
Author Response
Response to Reviewer 1
We’d like to express our appreciation to the Reviewer for the time and effort in revising our manuscript! Here we present our point-by-point answers to the suggestions and criticism.
- “Two sub-section have the same number 2.1.”
Answer: We thank the Reviewer to point out that editing mistake! It was rectified during the revision.
- “I would suggest to talk about the innate immunity in the first 2.1 sub-section since the innate immunity is deeply involved in the MS pathology and pathogenesis. Microglia is both detrimental and beneficial, it would be interesting to add few lines in the 2.1 sub-section. DOI: 10.3389/fncel.2020.00284.”
Answer: We thank the Reviewer for this suggestion. We expanded the “pathogenesis of MS” sub-section with information on the innate immune system’s role in the pathological processes leading to MS.
“Whichever process is the most important, the activation of all three type of lympho-cytes result in the recruitment and activation of microglia and macrophages, leading to the destruction of the myelin sheath and the appearance of the focal white matter lesions characteristic to the disease highlighting the importance of the innate immune system in MS as well [23]. These activated mononuclear phagocytes’ roles are mul-ti-layered, sometimes even opposing each other. On one hand, they are the effector cells of myelin destruction and removal, thus the primary cells responsible for the cre-ation of the focal white matter lesions characteristic to the disease [24]. The level of myelin degradation products in these cells is even considered to be a measure of activ-ity in these lesions [25]. On the other hand, however, several studies found that the “properly” conditioned microglia are able to promote remyelination as well. Induced by IL-4, microglia are able to promote the survival of oligodendrocytes and activate oligodendrocyte progenitors through several pathways (BDNF, TGF-β for example), thus play a pivotal role in the process of remission [26, 27]”
“Through these inflammatory responses – mainly microglia being the source of the oxidative stress,- the autoimmune process will inevitably lead to axonal dissection in the white matter and cortical and subcortical neuronal injury in the grey matter, evi-dently from the very beginning of the disease [31]”.
We again would like to thank all the suggestions and observations about our work to the Reviewer! We feel that the article became better with the suggested changes and additions. I hope that our revised manuscript will be acceptable for publication.
Yours sincerely,
Prof. Dr. László Vécsei, MD, D.Sc
Professor
Department of Neurology, University of Szeged
Reviewer 2 Report
This review is devoted to the role of proteomics in multiple sclerosis (MS) studies. The authors provide quite detailed information on MS pathogenesis, proteomics tools (maybe a little bit too detailed part), and on different aspects of proteomics studies performed in this field. Particular attention is devoted to identification of potential bio markers, and authors discuss ones in subsection of section 4.
Authors need to clarify better differences and similarities observed in proteomic studies performed in animal models, and in humans.
The main criticism.
The review needs in more illustrations. It would be nice to have one figure devoted to mechanisms of pathogenesis of MS, and, at least one figure devoted to main findings of proteomic studies (from section 4).
Minor criticisms.
List of abbreviations used would benefit the paper.
Lines 93-96. The sentence should be rewritten for clarity. The English language should be checked throughout the manuscript.
Line 110-123. The text should be formatted in a uniform way in whole manuscript.
Numbers of subsections of section 2 should be corrected (now all of them have number 2.1).
Figure 1. Section in Fig.1 called section analysis has Edman sequencing mention. I am not sure this method is widely used. It would be nice to have references showing research papers where this method was utilized (or it can be removed as a method which is not in active usage now). To a certain degree, mass-spectrometry can be attributed to sequence analysis as well since it can help with identification of proteins of interest (for example, using analysis of peptides obtained after trypsin treatment of a particular protein/s).
Line 259. Word expansive is probably a result of a typo.
Line 283. Despite these issues, biomarker research has exponentially intensified in the last two decades. This sentence should be checked for a grammatical error.
Correctness of references used should be checked throughout the text since, for example, in line 416 there is a mention of Mikkat et al, but there is no such author in corresponding reference number 97. There is no reference provided for Rafael et al in lines 448-449.
Table 1 is shown in the manuscript before it is mentioned in the text.
The review can be accepted after minor revision.
Author Response
Response to Reviewer 2
We’d like to express our appreciation to the Reviewer for the time and effort in revising our manuscript! Here we present our point-by-point answers to the suggestions and criticism.
The main criticism.
- “The review needs in more illustrations. It would be nice to have one figure devoted to mechanisms of pathogenesis of MS, and, at least one figure devoted to main findings of proteomic studies (from section 4).”
Answer: Thank you for the suggestion! A new figure about the pathogenesis of MS has been added to the manuscript (Figure 1), to make the article more illustrative. However, as the purpose of Table 1 is to summarize the most important findings of section 4, we felt it would be redundant to put another type of illustration in that manuscript about the same section. We hope the new Figure illustrates the text satisfactorily.
Minor criticisms.
- “List of abbreviations used would benefit the paper.”
Answer: Thank you for the suggestion! A list of abbreviations has been added at the end of the manuscript per your suggestion.
- “Lines 93-96. The sentence should be rewritten for clarity. The English language should be checked throughout the manuscript. “
Answer: The sentence was rephrased for clarification per the suggestion.
“Some promising data emerged implying that oligodendrocyte apoptosis and axonal damage, thus neurodegeneration is the first step in the evolution of MS. This process reveals previously hidden antigens and activates the immune system, leads to a secondary autoimmune response, initiating the whole “vicious circle”
- “Line 110-123. The text should be formatted in a uniform way in whole manuscript” and “Numbers of subsections of section 2 should be corrected (now all of them have number 2.1).
Answer: Thank you for the criticism. Both mistakes were made during the modification of formatting by the Editorial Office to their own standards. We corrected it best we could and will notify the Editorial Office.
- “Figure 1. Section in Fig.1 called section analysis has Edman sequencing mention. I am not sure this method is widely used. It would be nice to have references showing research papers where this method was utilized (or it can be removed as a method which is not in active usage now). To a certain degree, mass-spectrometry can be attributed to sequence analysis as well since it can help with identification of proteins of interest (for example, using analysis of peptides obtained after trypsin treatment of a particular protein/s)”.
Answer: Edman sequencing was removed from the text and the figure as well per your suggestion. However, we were not able to delete the old figure and insert the new into the text, thus, we will upload the new version of the figure as a separate file and ask the Editorial Office to change them.
- “Line 259. Word expansive is probably a result of a typo.”
Answer: Thank you, the typo was corrected!
- “Line 283. Despite these issues, biomarker research has exponentially intensified in the last two decades. This sentence should be checked for a grammatical error.“
Answer: The sentence was rephrased per your suggestion.
“Despite this, the number of researches aiming to identify biomarkers for neurological conditions has exponentially grew in the last two decades.”
- “Correctness of references used should be checked throughout the text since, for example, in line 416 there is a mention of Mikkat et al, but there is no such author in corresponding reference number 97. There is no reference provided for Rafael et al in lines 448-449.”
Answer: Thank you for discovering these mistakes! Both were rectified during the revision.
- “Table 1 is shown in the manuscript before it is mentioned in the text.”
Answer: Thank you for this criticism. We realize, that it is unusual to show a table before mentioning it in the text. However, as it is a comprehensive summary of Section 4, it could not have been mentioned earlier without that mention being awkward. Thus, it was first described in the Conclusions section. However, we felt, it better suits its purpose if it is immediately shown after Section 4, to summarize the many details described there and give it a perspective. We hope, we could give an acceptable explanation to this unconventional move, however, if the Reviewer still suggest the table to be moved, we will do so of course.
We again would like to thank all the suggestions and observations about our work to the Reviewer! We feel that the article became better with the suggested changes and additions. I hope that our revised manuscript will be acceptable for publication.
Yours sincerely,
Prof. Dr. László Vécsei, MD, D.Sc
Professor
Department of Neurology, University of Szeged
Reviewer 3 Report
The review submitted by the authors is of relevance to the area of neurodegeneration. The manuscript is well organized and written.
However, this reviewer found a low novelty in the data presented in the manuscript since the paper that was published by Monokesh K Sen et al (Int J Mol Sci . 2021 Jul 9;22(14):7377. doi: 10.3390/ijms22147377.) is recent and discuss the similar points. for example, in Sen's paper methods are deeply described compared to the current manuscript also including pitfalls and differences observed between animal and human models.
To improve the manuscript, the authors have to extend and expand the discussion from the data in the literature on the different body fluids. In particular, I suggest adding a section discussing how proteomics aids to detect biomarkers and mechanisms of disease progression compared to other methods. Overall, these suggestions should make a manuscript with a complementary scope to Sen's and add novelty for researchers in the area.
Author Response
Response to Reviewer 3
We’d like to express our appreciation to the Reviewer for the time and effort in revising our manuscript! Here we present our point-by-point answers to the suggestions and criticism.
- “To improve the manuscript, the authors have to extend and expand the discussion from the data in the literature on the different body fluids. In particular, I suggest adding a section discussing how proteomics aids to detect biomarkers and mechanisms of disease progression compared to other methods.”
Answer: Thank you for your suggestion! We added a paragraph detailing the advantages of proteomics as a method for biomarker research into the section.
“3.3. Why proteomics?
There are several attributes of proteomics that makes it a better choice for bi-omarker research than other methods. Despite great advances in DNA sequencing and genetics as a whole, it was found that almost 40% of protein-coding genes lack experi-mental evidence at the protein level [95]. Also, the variance between the mRNA levels and protein levels can reach 20-30-fold differences [96]. Additionally, post-translational modification (acetylation or glycolisation for example) can very much alter the function of the proteins and its role in disease development or progres-sion, which has no “sign” on the genomic or transcriptomic levels [97]. As pro-teins/peptides are the effector molecules in biological processes, these differences make the direct measurement of proteins inevitable.
We have elaborated on the several hardships in the identification of candidate protein biomarkers above - low abundance in easily collectable samples, the masking effect of resident proteins, or their fast degradation by protease enzymes after sample collection, to name a few. However, new technologies are continuously being devel-oped with success to overcome these issues. For example, Luchini et al reported the development of “smart” nanoparticles that can be immediately mixed with the col-lected sample and may perform chromatography and sequester the proteins in ques-tion away from albumin at the same time [98]. Despite this technology is still in the development phase and the costs are tremendous, the report do show, that the tech-nology of proteomics evolves at a tremendous speed, and the obstacles of the present will become solved in the near future.”
We again would like to thank all the suggestions and observations about our work to the Reviewer! We feel that the article became better with the suggested changes and additions. I hope that our revised manuscript will be acceptable for publication.
Yours sincerely,
Prof. Dr. László Vécsei, MD, D.Sc
Professor
Department of Neurology, University of Szeged